# Use of Plasmapheresis and Immunosuppressants to Treat Diffuse Alveolar Hemorrhage in a Patient with Granulomatosis with Polyangiitis

**DOI:** 10.3390/medicina55070378

**Published:** 2019-07-16

**Authors:** Yasar Sattar, Ammu Thampi Susheela, Waqas Ullah, Norina Usman, Fnu Zafrullah

**Affiliations:** 1Icahn School of Medicine at Mount Sinai-Elmhurst Hospital, Queens, NY 11373, USA; 2Beth Israel Deaconess Medical Center, Harvard Medical School, Boston, MA 02215, USA; 3Abington Hospital Jefferson Health, Abington, PA 19001, USA; 4Lasante Health Clinic, Brooklyn, NY 11226, USA; 5Steward Carney Hospital, Tufts Medical Center, Boston, MA 02111, USA

**Keywords:** diffuse alveolar hemorrhage, granulomatosis with polyangiitis, plasmapheresis, prednisone, cyclophosphamide, rituximab

## Abstract

Granulomatosis with polyangiitis (GPA) is a systemic granulomatous inflammatory disease characterized by small-to-medium vessel vasculitis due to Central Anti-Neutrophil Cytoplasmic Antibody (C-ANCA). GPA commonly involves the lungs and the kidneys. Among the pulmonary manifestations, diffuse alveolar hemorrhage (DHA) is a rare presentation of GPA that can present with hemoptysis leading to acute onset of anemia and hemodynamic instability. An active diagnostic workup including serologic titer of C-ANCA, imaging, intensive care, and aggressive immunosuppression is the key to DAH management. We report a case of DAH secondary to GPA that presented with hemoptysis leading to severe anemia, initially resuscitated symptomatically and started on plasmapheresis with pulse steroids and cyclophosphamide. Timely diagnosis and management led to a remarkable recovery of the pulmonary symptoms and imaging findings of DAH.

## 1. Introduction

Granulomatosis with polyangiitis (GPA) is a chronic granulomatous small-medium-vessel necrotizing inflammation of the lungs and kidneys. It is one of the three antineutrophil cytoplasmic antibody-associated vasculitides (AAV), the others being microscopic polyangiitis (MPA) and eosinophilic granulomatosis with polyangiitis (EGPA) [1]. GPA is extremely rare in the United States, with an incidence of only 8–10 cases per 1 million residents. The peak incidence of GPA is during the sixth and seventh decades of life. It may involve any organ, including the lungs, kidney, heart, skin, joints, and eyes [2]. The lung involvement can present as asymptomatic pulmonary nodules but, in rare cases, can cause diffuse alveolar hemorrhage (DAH) that can present as massive hemoptysis leading to hemodynamic decompensation. DAH can cause up to 60% mortality in GPA. The diagnosis is challenging, given the nonuniform and nonspecific symptoms. DAH can be diagnosed with a combination of pulmonary symptoms, positive serology, imaging, lung biopsy, and a flexible fiberoptic bronchoscopy lavage with persistent bloody aliquots [3]. A flexible fiberoptic bronchoscopy is the most definitive test for the diagnosis of DAH. Here, we present the case of a 60-year-old female with DAH secondary to GPA that was treated with plasmapheresis, pulse steroids, and cyclophosphamide. 

## 2. Case Presentation

A 60-year-old female with a medical history of unexplained prolonged hematuria for the last four years presented to the emergency department (ED) with complaints of hemoptysis, fever, myalgia, and night sweats that had persisted for at least one week. She had no other comorbidities, and her family history was unremarkable. Before coming to the ED, the patient experienced four episodes of hemoptysis and was referred from urgent care to the ED due to the concern of acute anemia and chronic kidney injury. The patient gave her informed consent for inclusion before participating in the study.

On admission, vital signs included blood pressure of 162/93 mmHg, heart rate of 98 beats/minute, respiratory rate of 24 breaths/minute, and oxygen saturation of 94%. A physical examination yielded non-significant findings except for pallor, blood around the nares, and prominent right-sided pulmonary crackles on auscultation. 

The results of the initial laboratory tests, including hematology, metabolic panel, arterial blood gas, and culture and sensitivity, are shown below in Table 1.

Urine analysis and baseline antibody panel for diagnostic workup are shown in Table 2 and Table 3 respectively. 

A baseline chest radiography finding is shown in Figure 1. A high-resolution computed tomography (HRCT) of the chest is shown below in Figure 2. 

A flexible fiberoptic bronchoscope was introduced through the endotracheal tube and advanced to the tracheobronchial tree of both lungs under adequate sedation. Fresh blood was found throughout the tracheobronchial tree, with no focal bleeding focus. During serial aliquoting, the blood remained unchanged, as aliquots progressed with 60 mL saline flushes.

A bronchoscopy image showed blood in the right middle lobe in the tracheobronchial tree, failing to show the specific focal source, as shown in Figure 3. 

A microscopic evaluation of the broncho-alveolar lavage showed a red color aspirate with numerous pulmonary alveolar macrophages and red blood cells; bacterial and fungal cultures, as well as acid-fast staining of the broncho-alveolar aspirate were negative. 

The auto-antibody panel indicated a very high level of C-ANCA. The initial laboratory tests revealed a low estimated glomerular filtration rate and a high creatinine level, as well as an increase in anion gap, metabolic acidosis, hyperphosphatemia, parathyroid hormone level, and a decrease of vitamin D level. The nephrology department was consulted, and a kidney biopsy was performed. The renal biopsy findings suggested pauci-immune (ANCA-associated), rapidly progressive glomerulonephritis. Specifically, light microscopy revealed a medullary segment with no cortex or glomeruli and intraluminal hyaline casts. Cortical sections contained three glomeruli containing fibro-cellular crescents. Electron microscopy revealed wrinkled glomerular basement membranes and capillary luminal compression with variable levels of endothelial cells and diffused effacement of the foot processes. Additionally, scattered electron-dense mesangial deposits and a loss of brush borders on adjacent tubules were observed. Immunofluorescence staining was inconclusive. A presumptive diagnosis of diffuse alveolar hemorrhage due to GPA was made based on the constellation of anemia, positive C-ANCA test, high proteinase-3 antibody titer, bronchoscopy findings of diffuse alveolar hemorrhage without a focal source, chronic kidney disease stage-5, and renal biopsy findings of rapidly progressive glomerulonephritis. Other differential diagnoses included:(A)Microscopic polyangiitis—this was difficult to differentiate on the basis of the clinical background, but a negative myeloperoxidase titer helped us to rule this out.(B)Eosinophilic granulomatosis with polyangiitis—this was difficult to differentiate on the basis of the clinical presentation, but this was ruled out considering the negative myeloperoxidase findings and a normal eosinophil count.(C)Drug-induced vasculitis—although this can have a similar presentation, a thorough history-taking ruled out this disease.

We administered a seven-day course of plasmapheresis on alternate days together with a combination of pulse steroids (methylprednisolone 1000 mg twice daily) for seven days and cyclophosphamide 730 mg q3/week. Antibiotic coverage with vancomycin 1000 mg and ceftriaxone 1000 mg was also given during plasmapheresis and immunosuppressants. During treatment, the respiratory status was supported by low tidal volume minute ventilation followed by successful extubation. For chronic kidney disease (CKD) treatment, the patient underwent hemodialysis through a right subclavian permanent catheter with a transition to the arteriovenous fistula route for an outpatient dialysis route. 

A multidisciplinary therapeutic approach including fluid support, ventilatory support, plasmapheresis, pulse steroids, and cyclophosphamide led to a remarkable symptomatic improvement of the pulmonary symptoms and to the resolution of the pulmonary imaging findings, as shown in part B of Figure 1. 

The patient was discharged with outpatient prednisone 40 mg with a taper over four weeks. For end-stage renal disease secondary to granulomatosis with polyangiitis, our patient received scheduled outpatient hemodialysis through an arteriovenous (AV) fistula.

## 3. Discussion

GPA is a chronic systemic disorder involving necrotizing inflammation of small-to-medium vessels that may affect any organ. GPA may present with various signs and symptoms, including nonspecific fever, arthralgia, and malaise, and may involve fatal cardiac, pulmonary, and renal manifestations [4]. GPA diagnostic criteria include the American College criterion and the Chapel Hill Consensus criterion.

According to the American College of Rheumatology criteria, a diagnosis of GPA requires two of the following four conditions. 

Nasal or oral inflammation (e.g., painful or painless oral ulcers or purulent or bloody nasal discharge).Abnormal chest radiography scans showing nodules, fixed infiltrates, or cavities.Abnormal urine analysis (microscopic hematuria with or without red cell casts).Granulomatosis inflammation upon biopsy of an artery or perivascular area.

The Chapel Hill Consensus Criterion (CHCC) is usually not used for diagnostic purposes but is generally used for definitions. The CHCC criteria include the size of the affected vessels and the organ involved. 

Large-vessel vasculitis includes Takayasu and Giant cell arteritis.Medium-vessel vasculitis includes polyarteritis nodosa and Kawasaki disease.Small-vessel vasculitis includes AAV, MPA, GPA, EGPA, immune complex vasculitis, anti-glomerular basement membrane disease, cryoglobulinemic vasculitis, IgA vasculitis, and hypocomplementemia urticarial vasculitis.Variable-vessel vasculitis includes Behcet’s syndrome and Cogan’s syndrome.Single-organ vasculitis includes vasculitis in only one organ with no systemic manifestation.Vasculitis associated with systemic disease includes vasculitis secondary to systemic disease, comprising rheumatoid arthritis, lupus vasculitis, etc.Vasculitis associated with probable etiology, including hepatitis B, hepatitis C, and hydralazine.

The CHCC also helps exclude some AAV vasculitis with the following features:The absence of immune complex deposits, differentiating small vessel vasculitis from GPA, MPA, and EGPA.The presence of necrotizing vasculitis without granuloma, distinguishing MPA from EGPA and GPA.
The potential value of ANCA levels are not included as diagnostic criteria.

The lung is a common site of GPA, and DAH is a severe manifestation of this disease that affects up to 10% of patients. DAH can present dyspnea/cough with hypoxemia, hemoptysis leading to progressive anemia, and bronchoalveolar lavage showing blood in a diagnosed case of GPA [5,6]. 

The diagnosis of DAH secondary to GPA can be made by basic laboratory tests, including complete blood count (CBC), urine analysis, serology, imaging, biopsy, and bronchoalveolar lavage findings. The CBC can show non-specific elevations of white blood cells, platelet counts, C-reactive protein (CRP) level, and erythrocyte sedimentation rate. Urine analysis can show RBC cast, hematuria, and even signs of chronic kidney disease, like fatty wax cast. Serology findings include a high C-ANCA titer with an antiproteinase-3 pattern, as seen in our patient. Chest radiography and HRCT can reveal pulmonary nodules, ground-glass opacities, and infiltrates. Biopsy findings are a cornerstone in the diagnosis of DAH secondary to GPA. A renal biopsy can show features consistent with GPA, including rapidly progressive glomerulonephritis (RPGN) with or without granuloma. In the cases of RPGN without granuloma, lung and skin biopsies can also aid in the diagnostic workup of GPA. Skin biopsies are preferred over lung biopsy, given their less invasive nature. Bronchoalveolar lavage (BAL) with a flexible fiberoptic bronchoscope is the gold standard test in the diagnostic workup of DAH. In acute cases, BAL can show serial blood on three sequential aliquots without any specific focus or, sometimes, from a single-point focus. In chronic and recurrent cases of DAH, BAL can show hemosiderin-laden macrophages with >20% siderophages out of total pulmonary macrophages in the specimen. This phenomenon of blood in alveoli with capillaritis in DAH can also increase the diffuse capacity of carbon monoxide up to 30% from baseline [7,8]. 

Before making a diagnosis of GPA, other primary vasculitides causing DAH should be ruled out by the absence of perinuclear-ANCA and anti-glomerular basement membrane antibodies. Infectious etiologies, such as hepatitis B and C viruses, human immunodeficiency virus, and pulmonary tuberculosis, as well as secondary causes of vasculitis with similar presentations, including infectious, drug-related, and other autoimmune diseases, must also be ruled out. As drugs such as hydralazine and propylthiouracil can also cause ANCA vasculitis, extensive history-taking can help to rule out drug-induced vasculitis with the differential diagnosis. In rare case, angiosarcoma of the pulmonary vessels can also cause DAH and should be considered in differentials [8]. 

The pathophysiology of DAH in GPA is attributed to pulmonary vasculitis and uremia-induced platelet dysfunction, which increases alveolar bleeding [9]. Pulmonary vasculitis can affect alveolar capillaries and venules, likely secondary to antibody-mediated vessel destruction. Uremia-induced platelet dysfunction increases the bleeding risk in GPA patients and can also be a potential risk of DAH. 

The treatment of DAH secondary to GPA includes remission followed by maintenance therapy. The initial management includes hemodynamically stabilization of DAH. After hemodynamic stabilization in severe DAH, remission should be achieved with a combination of plasmapheresis and immunosuppressants like pulse steroids and high-dose cyclophosphamide. If patients are not responsive to steroids or cyclophosphamide, other alternative immunosuppressants, including azathioprine, mycophenolate mofetil, etanercept, methotrexate, and rituximab, should be administered [1,10]. The excellent efficacy of plasmapheresis was demonstrated in a retrospective study by Klemmer et al. In this study, 20 patients received a combination of plasmapheresis with intravenous immunosuppressive therapy and showed a 100% remission rate for pulmonary symptoms with some improvement of renal function [3]. Another study by Frasca et al. reported that the early administration of 3–10 plasmapheresis sessions together with immunosuppressive therapy significantly reduced ANCA levels, which slowed the progression to end-stage renal disease and mortality [11]. In a study conducted by Cartin-Ceba et al, out of the 73 patient who had plasma exchange, 32 were not associated with remission after 6 months (OR) of 0.49 (95% CI 0.12–1.95) *p* = 0.32, whereas patients receiving Rituximab were associated with remission at 6 months (OR 6.45, 95% CI 1.78–29), *p* = 0.003 [12]. Thus, the therapeutic indication of plasma exchange and glucocorticoids for the treatment of AAV remains controversial (PEXIVAS-NCT00987389). The results, so far, do not show a mortality benefit for plasma exchange in patients with DAH [13]. Plasma exchange is also associated with an increased risk of bleeding, and hence, citrate is used as an anticoagulant to reduce the complications [14,15].

In the present case, our patient improved after seven plasmapheresis sessions combined with cyclophosphamide and methylprednisone. However, the patient’s renal function did not return to normal due to the stage-5 CKD. Therefore, early diagnosis and management of GPA and DAH showed a significant recovery of the pulmonary symptoms, with limited to no improvement of the renal manifestation. 

## 4. Conclusions

Physicians should include DAH as a cause of massive hemoptysis in patients with granulomatosis with polyangiitis (GPA). DAH diagnosis relies on a combination of clinical, serologic, imaging, and biopsy findings with bronchoalveolar lavage. The treatment of severe DAH with massive hemoptysis can be started with multiple sessions of plasmapheresis, pulsed high-dose steroids (1 g daily for three days and 1 mg/kg daily thereafter,) and cyclophosphamide therapy (1.2 g/pulse dose every two weeks for the first three pulsed doses followed by infusion every third week for the next two pulsed treatments). A combination of plasmapheresis with immunosuppressants reverses the pulmonary symptoms and imaging findings, but a reversal of renal disease is limited.

## Figures and Tables

**Figure 1 medicina-55-00378-f001:**
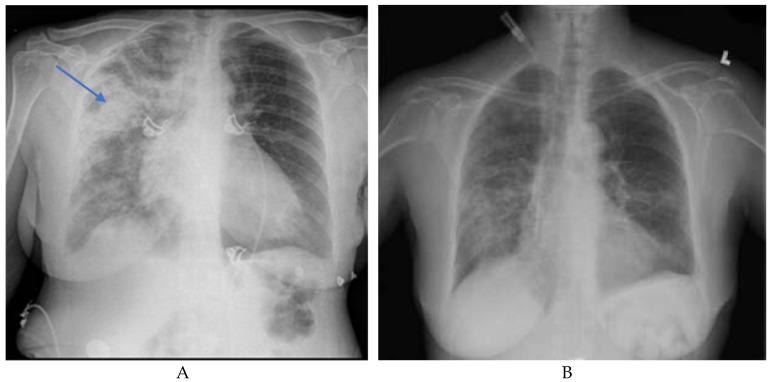
(**A**) Baseline chest radiography showing dense alveolar infiltration in the right upper lung (blue arrow). (**B**) Post-treatment chest radiography showing clear lung fields and decreased alveolar infiltration status.

**Figure 2 medicina-55-00378-f002:**
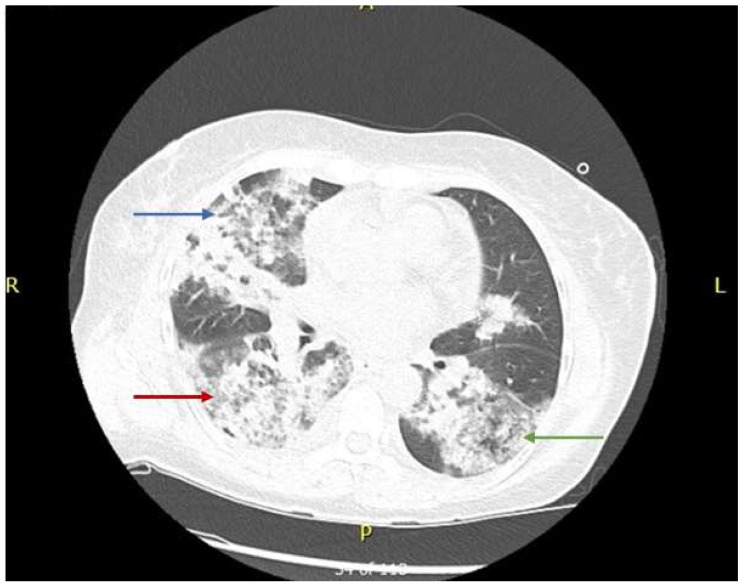
High-resolution computed tomography (HRCT) scan showing a nodular cavity with infiltration in the right upper (red arrow), middle lobe (blue arrow), and left upper lobe infiltrates (green arrow).

**Figure 3 medicina-55-00378-f003:**
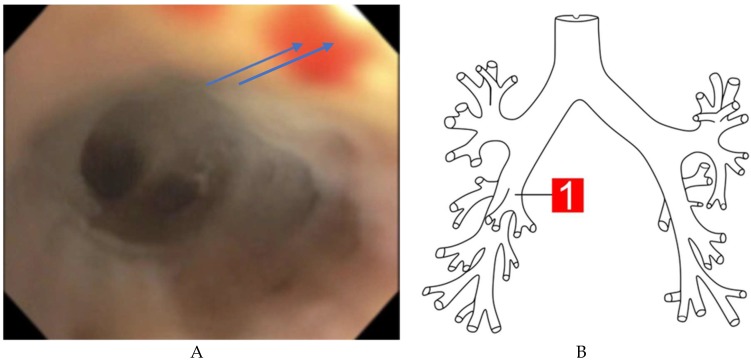
(**A**) Bronchoscopy showing blood (blue arrow) in the right middle lobe in the tracheobronchial tree. (**B**) Site of blood shown in the bronchoscopy image.

**Table 1 medicina-55-00378-t001:** Baseline laboratory values.

Analyte	Baseline	Reference Range
**Hematology**		
WBC	21.8 K/mcL	4.5–11.0 K/mcL
Hemoglobin	5.8 gm/dL	12.0–16.0 gm/dL
RBC	2.10 M/mcL	4.00–5.20 M/mcL
Platelet count	356 K/uL	550 K/uL
Neutrophils	90.5%	43.0–77.0%
Lymphocytes	3.8%	13.0–44.0%
Monocyte	6.2%	2.0–14.0%
Eosinophil	0.1%	0.0–6.0%
Basophil	0.5%	0.0–2.0%
MCV	90.1 fl	80.0–100.0 fl
MCH	27.8 pg	27.0–34.0 pg
MCHC	30.8 gm/dL	32.0–36.0 gm/dL
RDW	17.4%	10.3–14.5%
PT	13.0 seconds	10.0–13.0 seconds
aPTT	27.4 seconds	27.0–36.0 seconds
**Metabolic panel**		
ALT	7 U/L	7–35 U/L
AST	11 U/L	10–35 U/L
Total Bilirubin	0.3 mg/dL	0.1–1.2 mg/dL
BNP	616 pg/mL	1–100 pg/mL
C-reactive protein	178.3 mg/L	0.0–1.0 mg/L
ESR	116 mm/hr	0–20 mm/hr
Creatinine	8.83 mg/dL	0.40–1.60 mg/dL
BUN	70 mg/dL	8–22 mg/dL
e-GFR	5 ml/min/1.73 m^2^	90–120 ml/min/1.73 m^2^
Phosphorus	7.4 mg/dL	2.5–4.9 mg/dL
Calcium	8.3	8.5–10.5 mg/dL
Sodium	136 mEq/L	136–146 mEq/L
Vitamin D 25- Hydroxy	22.6 ng/mL	30.0–100.0 ng/mL
PTH	160.5 pg/mL	10.0–65.0 pg/mL
**Arterial Blood Gases**		
pH	7.46	7.320–7.420
pCO_2_	31.2 mmHg	38.0–50.0 mmHg
HCO_3_	23.6 mmol/L	22.0–28.0 mmol/L
Anion gap	17 mmol/L	2–11 mmol/L
Lactate	1.0 mmol/L	0.0–2.2 mmol/L
**Culture and sensitivity**		
Sputum AFB	Negative in three samples	Negative
Stool culture	Negative	Negative
Urine culture	Negative	Negative
Blood culture	Negative	Negative

Abbreviations: WBC: white blood cells; RBC, red blood cells; MCV, mean corpuscular volume; MCH, mean corpuscular hemoglobin; MCHC, mean corpuscular hemoglobin concentration; RDW, red blood cell distribution width; PT, prothrombin time; aPTT, activated partial thromboplastin time; ALT, alanine aminotransferase; AST, aspartate aminotransferase; BNP, brain natriuretic peptide; ESR, erythrocyte sedimentation rate; BUN, blood urea nitrogen; e-GFR, estimated glomerular filtration rate; PTH, parathyroid hormone; pCO2, partial pressure of carbon dioxide; HCO3, bicarbonate; AFB, acid-fast bacillus.

**Table 2 medicina-55-00378-t002:** Detailed urine analysis with reflex microscopy.

Analyte	Results
Appearance	Cloudy (Reference: clear)
Color	Yellow (Reference: yellow)
Urine pH	6.0 (Reference: 5.0–7.5)
Specific gravity	1.016 (Reference: 1.005–1.030)
Bilirubin	Negative (Reference: negative)
Ketones	Negative (Reference: negative)
Glucose	Negative (Reference: negative)
Blood	Large (Reference: negative)
Protein	1000 mg/dL (Reference: negative mg/dL)
Urobilinogen	1.0 (Reference: 0.2–1.0)
Nitrite	Negative (Reference: negative)
Leukocyte esterase	Negative (Reference: negative)
White blood cells urine	>50 HPF (Reference: 0–4 HPF)
Red blood cells urine	<50 HPF (Reference: 0–3 HPF)
Urine bacteria	Negative (Reference: negative)
Squamous epithelial cells	0–4 HPF (Reference: 0–4 HPF)
Casts	None (Reference: no cast)

**Table 3 medicina-55-00378-t003:** Baseline autoimmune panel.

Analyte	Result	Reference
ANA antibody	1:160	1:180
ANA pattern	Homogenous	
Anti-GBM antibodies	Negative (<0.2 U)	Negative: <1.0 U Positive: >1.0 U
C-ANCA	Positive (1:320 titer)	
Proteinase-3 antibody	Positive (>150.0 units)	Negative: <20 units Diffusely positive: 21.0–30.0 units Positive: >30 units
Myeloperoxidase antibody	Negative (5.1 units)	Negative: <20 units Diffusely positive: 21.0–30.0 units Positive: >30.0 units
Anti-U1RNP Antibody	Negative (0.4 AI)	Negative: <1.0 AI Positive: ≥1.0 AI
Anti SCL-70	Negative (<0.2 AI)	Positive: ≥1.0 AI
ASO	Negative	
Anti-DsDNA	Negative (<12 IU/mL)	Positive: >75 IU/mL
Hepatitis C Antibody	Non-reactive	
Hepatitis B Antibody	Non-reactive (<10 mIU/mL)	Reactive: ≥10 mIU/mL
C3	113 mg/dL	81–157 mg/dL
C4	25 mg/dL	13–39 mg/dL
Rheumatoid factor	22 IU/mL	0–13 IU/mL
Anti-SS-A-AB(RO)	Negative; <0.2 AI	Negative: <1.0 AI Positive: >1.0 AI
Cyclic Citrullinated C-Peptide Ab	<8 U	Negative: ≤39 U Positive: >40 U

Abbreviations: ANA, anti-nuclear antibodies; GBM, glomerular basement membrane; C-ANCA, cytoplasmic anti-neutrophilic antibodies; U1RNP, U1-ribonucleoprotein; SCL-70, scleroderma-70; ASO, anti-streptolysin O; DsDNA, double-stranded DNA; C3 and C4, complement components 3 and 4. Baseline chest radiography and status post treatment are shown in part A and B of Figure 1, respectively.

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
