# Peer review of "Use of Plasmapheresis and Immunosuppressants to Treat Diffuse Alveolar Hemorrhage in a Patient with Granulomatosis with Polyangiitis"

_medicina, 2019, doi:10.3390/medicina55070378_

Round 1

Reviewer 1 Report

Overall, this is an interesting topic not presented in a cohesive way. In its present state, it is very difficult to read.

1. The abstract needs better organization. For example, the abstract opens with the following sentence “Granulomatosis with polyangiitis (GPA) is a systemic granulomatous inflammatory disease characterized by Central Anti-Neutrophil Cytoplasmic Antibody (CANCA) causing small- to-medium vessel vasculitis” which is acceptable but then abruptly followed by “However, hemoptysis due to underlying diffuse alveolar hemorrhage is a rare presentation”. While an abstract is limited in scope, these two lines almost seem mutually exclusive. You need to link every sentence in an abstract for a smooth flowing succinct summary of your case which the abstract in the very first two lines fails to achieve.

2. I have two concerns with the introduction:

a. It is mentioned that DAH can be diagnosed with the combination of symptoms, positive serology, imaging and biopsy findings. In clinical practice the most common method of diagnosing DAH is with bronchoscopy with lavage demonstrating persistent bloody aliquots. This is mentioned in the case presentation but is not represented in this line which is only partially accurate. 

b. Please delete the phrase “lethal hemoptysis”. Hemoptysis can be categorized as massive/life threating hemoptysis or not. The term lethal hemoptysis is likely used to indicate that the patient did not survive but is not medically accepted jargon.

3. The case presentation serves us a plethora of laboratory data in a paragraph that is very, very difficult to read. This would be better presented in tables grouped into the specific biochemistry analysis e.g. hematology, metabolic panel, autoimmune, etc. Please note that the correct terminology if Flexible Fiberoptic Bronchoscopy and not Flexible Bronchoscopy. The last paragraph of the case presentation is very poorly written and reflects poor sentence structure. For example: “Arterio-venous (AV) fistula was performed by a member of the vascular surgery department for permanent vascular access. Patient respiratory status becomes medically optimized.” These are two phrases joined together to provide an illusion of prose.

4. It was very difficult for me to get through the discussion as there were no paragraphs except for the conclusion. Paragraphs in a discussion represents separate thoughts on disease presentation, diagnosis and treatment and reflect current literature and then related to the described patient. This is done in poor fashion.

In its present state, the article needs to be completely re-written

Author Response

1. The abstract needs better organization. For example, the abstract opens with the following sentence “Granulomatosis with polyangiitis (GPA) is a systemic granulomatous inflammatory disease characterized by Central Anti-Neutrophil Cytoplasmic Antibody (CANCA) causing small- to-medium vessel vasculitis” which is acceptable but then abruptly followed by “However, hemoptysis due to underlying diffuse alveolar hemorrhage is a rare presentation”. While an abstract is limited in scope, these two lines almost seem mutually exclusive. You need to link every sentence in an abstract for a smooth flowing succinct summary of your case which the abstract in the very first two lines fails to achieve. Response: Thank you for this insightful observation. We re-wrote the abstract as following “Granulomatosis with polyangiitis (GPA) is a systemic granulomatous inflammatory disease characterized by small-to-medium vessel vasculitis due to Central Anti-Neutrophil Cytoplasmic Antibody (C-ANCA). GPA commonly involves lung and kidneys. Among pulmonary manifestation, diffuse alveolar hemorrhage is a rare presentation of GPA that can present with hemoptysis leading to acute onset of anemia and hemodynamic instability. An active diagnostic workup, including serologic titer of C-ANCA, imaging, intensive care, and aggressive immunosuppression is the key to DAH management. We reported a case of DAH secondary to GPA presented with hemoptysis leading to severe anemia, initially resuscitated symptomatically, and started on plasmapheresis with pulse steroids and cyclophosphamide. Timely diagnosis and management lead to a remarkable recovery of pulmonary symptoms and imaging findings of DAH.” 2. I have two concerns with the introduction: a. It is mentioned that DAH can be diagnosed with the combination of symptoms, positive serology, imaging and biopsy findings. In clinical practice the most common method of diagnosing DAH is with bronchoscopy with lavage demonstrating persistent bloody aliquots. This is mentioned in the case presentation but is not represented in this line which is only partially accurate. b. Please delete the phrase “lethal hemoptysis”. Hemoptysis can be categorized as massive/life threating hemoptysis or not. The term lethal hemoptysis is likely used to indicate that the patient did not survive but is not medically accepted jargon. Response: Thank you for this excellent review. We added the bronchoscopy test in diagnosis in the introduction section. We removed the word “lethal hemoptysis”. We re-wrote the whole introduction as “Granulomatosis with polyangiitis (GPA) is a chronic granulomatous small-medium vessel necrotizing inflammation of the lungs and kidneys. It is one of the three antineutrophil cytoplasmic antibody-associated vasculitides (ANCA), others being microscopic polyangiitis and eosinophilic granulomatosis with polyangiitis. [1] GPA is extremely rare in the United States, with an incidence of only 8–10 cases per 1 million residents. The peak incidence of GPA is during the sixth and seventh decades of life. It may involve any organ, including the lungs, kidney, heart, skin, joints, and eyes [2]. The lung involvement can present as asymptomatic pulmonary nodules, but in rare cases, it can cause diffuse alveolar hemorrhage (DAH) that can present as massive hemoptysis leading to hemodynamic decompensation. DAH can have mortality up to 60% in GPA. The diagnosis is challenging and demanding, given the nonuniform and nonspecific symptoms. DAH diagnosis made by a combination of pulmonary symptoms, positive serology, imaging, lung biopsy, and flexible fiberoptic bronchoscopy lavage with persistent bloody aliquots. [3] Flexible fiberoptic bronchoscopy is the most definitive test for the diagnosis of diffuse alveolar hemorrhage. Below we presented a case of a 60-year-old female with DAH secondary to GPA and treated with plasmapheresis, pulse steroids, and cyclophosphamide.” 3. The case presentation serves us a plethora of laboratory data in a paragraph that is very, very difficult to read. This would be better presented in tables grouped into the specific biochemistry analysis e.g. hematology, metabolic panel, autoimmune, etc. Please note that the correct terminology if Flexible Fiberoptic Bronchoscopy and not Flexible Bronchoscopy. The last paragraph of the case presentation is very poorly written and reflects poor sentence structure. For example: “Arterio-venous (AV) fistula was performed by a member of the vascular surgery department for permanent vascular access. Patient respiratory status becomes medically optimized.” These are two phrases joined together to provide an illusion of prose. Response: Thank you for highlighting this point. We added all the laboratory data separately in tables including hematology, metabolic, autoimmune, culture, urine analysis. The correct terminology of Flexible Fiberoptic Bronchoscopy has been added in the case presentation and introduction part also. The last paragraph of case presentation is re-written as: “For CKD, the patient underwent hemodialysis through a right subclavian permanent catheter with a transition to arteriovenous fistula route for an outpatient dialysis route. A multidisciplinary therapeutic approach including fluid support, ventilatory support, plasmapheresis, pulse steroids, and cyclophosphamide led to a remarkable symptomatic improvement of pulmonary symptoms and resolution of pulmonary imaging findings as shown in part B of figure 1. The patient was discharged with outpatient prednisone 40 mg with a taper over four weeks. For end-stage renal disease secondary to granulomatosis with polyangiitis, our patient received scheduled outpatient hemodialysis through AV fistula.” 4. It was very difficult for me to get through the discussion as there were no paragraphs except for the conclusion. Paragraphs in a discussion represents separate thoughts on disease presentation, diagnosis and treatment and reflect current literature and then related to the described patient. This is done in poor fashion. Response: Thank you for the excellent suggestion. We added separate paragraph now and we re-wrote the whole discussion section. Discussion section shows GPA definitions, symptoms, criteria, DAH symptoms, diagnosis, pathophysiology, treatment. Conclusion is also re-written. In its present state, the article needs to be completely re-written Response: Thank you for the excellent reviewer comments. We almost completely re-wrote the manuscript with extensive re-edits.

Reviewer 2 Report

Thank you for the opportunity to review this interesting case of diffuse alveolar hemorrhage due to granulomatosis with polyangiitis. The structure of paper is well organized. The diagnosis of GPA is absolutely confirmed by the presentation of hemoptysis, hematuria, findings of biomarker, bronchoalveolar lavage and renal biopsy.

Treatment response with plasmapheresis and immunosuppressants is also good in this patient. Therefore, this is a good case with education meaning to remind the physicians in dealing with diffused alveolar hemorrhage.

I suggest minor revision of this paper.

I have some comments: 

1. Could you improve the quality of Figure 1 and Figure 2.

2. Add some Arrow markers to indicate the lesions on Figure 1 and 2

3. If possible, could you provide the pathology findings of renal biopsy to show the renal involvement in this case. (I think this image is more important than the picture of bronchoscopic findings. You have made a very clear descriptions of BAL findings in Line 75-77)

Author Response

1.     Could you improve the quality of Figure 1 and Figure 2.

Response: Thank you for highlighting this point. A new high quality figures are added with an arrows demonstrating pathology.

2.     Add some Arrow markers to indicate the lesions on Figure 1 and 2.

Response: Thank you for highlighting this point. A new high quality figures are added with an arrows demonstrating pathology.

3.     If possible, could you provide the pathology findings of renal biopsy to show the renal involvement in this case. (I think this image is more important than the picture of bronchoscopic findings. You have made a very clear descriptions of BAL findings in Line 75-77).

Response: Thank you for highlighting this point. I added a new high quality image for bronchoscopy. Unfortunately, we do not have access to renal biopsy image as we only have report. At our hospital we sent sample to another hospital for surgical pathology review so it won’t be accessible. But we made a clear description of renal biopsy results.

4.     References

Response: We have fixed all the references per journal format and also included all the DOI’s.

Round 2

Reviewer 1 Report

The re-written manuscript reads well and I like the flow